# Influenza Vaccine Uptake in the United States before and during the COVID-19 Pandemic

**DOI:** 10.3390/vaccines10101610

**Published:** 2022-09-26

**Authors:** Ian McGovern, Alina Bogdanov, Katherine Cappell, Sam Whipple, Mendel Haag

**Affiliations:** 1Seqirus Inc., Cambridge, MA 02139, USA; 2Veradigm, San Francisco, CA 94103, USA; 3Seqirus NL BV, 1105 Amsterdam, The Netherlands

**Keywords:** influenza vaccine, COVID-19, electronic medical records, administrative claims, vaccination coverage, vaccine uptake, influenza seasons

## Abstract

The COVID-19 pandemic, along with disruptions to routine medical care, brought renewed urgency to public health messaging about the importance of influenza vaccination. This retrospective cohort study used a database of linked claims and electronic medical record data to evaluate clinical and demographic characteristics and influenza vaccination history associated with changes in influenza vaccine uptake following the start of the COVID-19 pandemic. Influenza vaccine uptake was examined in six seasons (2015–2016 through 2020–2021). Individuals were grouped by vaccination history in the five seasons before 2020–2021. Characteristics of 2020–2021 vaccinated vs. unvaccinated individuals were compared, stratified by vaccination history. Overall influenza vaccination uptake was highest in 2020–2021 (35.4%), following a trend of increasing uptake since 2016–2017 (31.4%). Uptake in 2020–2021 was observed in all age groups except ≥65 years, and the increase was particularly notable in individuals <18 years. In the previous five seasons, individuals ≤17 and >65 years, White, and Asian individuals were most likely, while 18-to-49-year-olds and those with fewer comorbidities were least likely, to be consistently vaccinated. Influenza vaccination status in 2020–2021 aligned with vaccination history; few differences in patient characteristics (age, comorbidities, state of residence) were observed when stratified by vaccination history.

## 1. Introduction

Influenza places a substantial burden on healthcare systems and communities. During the last three influenza seasons from 2017–2018 to 2019–2020, influenza led to 29 to 41 million illnesses, 16 to 17 million medical visits, 380,000 to 710,000 hospitalizations, and 20,000 to 52,000 deaths in the United States each season [1]. While influenza vaccination is important for every influenza season, there has been added emphasis on influenza vaccination during the coronavirus disease 2019 (COVID-19) pandemic. Since its declaration as a pandemic in March 2020 [2], as of 14 June 2022, COVID-19 infection has led to 86 million cases and over 1 million deaths in the United States as well as 536 million cases and 6.3 million deaths globally [3]. A higher level of influenza vaccine uptake is aimed to lessen the compounding of illness in patients and stress on the healthcare system caused by the cocirculation of both the severe acute respiratory syndrome coronavirus 2 (SARS-CoV-2) and influenza virus. The Centers for Disease Control and Prevention (CDC) set the target of vaccinating 65% of American adults for the 2020–2021 influenza season [4]. 

The increased public health messaging around the importance of influenza vaccination and disruptions to healthcare during the COVID-19 pandemic may result in changes to influenza vaccine uptake. A key factor in influenza vaccine uptake is a history of vaccinations [5,6,7]. Determining characteristics of 2020–2021 influenza vaccine recipients vs. nonrecipients stratified by prior vaccination history may inform specific population subsets who may benefit from increased information campaigns, improved vaccine access, or other public health interventions to improve influenza vaccine uptake. 

The objectives of this study are to (i) describe influenza vaccine uptake patterns during influenza vaccination seasons before (2015–2016 to 2019–2020) and during the COVID-19 pandemic (2020 to 2021) in an insured population; (ii) describe patient characteristics stratified by influenza vaccination history; and (iii) evaluate the characteristics of influenza vaccine recipients during the 2020–2021 season compared with nonrecipients according to their vaccination history.

## 2. Materials and Methods

### 2.1. Study Design

This was a retrospective observational study. Part 1 consisted of multiple season-specific, cross-sectional evaluations of influenza vaccine uptake overall and among specified subgroups from the 2015–2016 through the 2020–2021 vaccination seasons. Part 2 entailed the use of a multi-season cohort that was continuously enrolled in the linked dataset to determine influenza vaccination history in the previous five seasons. We evaluated overall patient characteristics within vaccination history groups as well as patient characteristics stratified by both vaccination history and influenza vaccination status during the 2020–2021 season.

### 2.2. Data Source

We used a linked dataset, which contained U.S. electronic medical records (EMRs) from Veradigm^®^ (namely, the Veradigm^®^ Health Insights Ambulatory EHR Database) and associated healthcare claims data from Komodo Health. The Linked Dataset integrates healthcare information of patients who have interacted with a physician practice indicated by the Veradigm^®^ EMR system and who have claims in the Komodo Health claims dataset. Medical and pharmacy claims data feeds began in 2015. The study used closed claims data, which originated from the payer and provided a comprehensive view of patient healthcare interactions. The dataset used comprised healthcare practices covering every 3-digit zip code in the United States. This study included data from six influenza seasons (2015–2016 through 2020–2021). The combination of national representativeness, large sample size, and comprehensive insights into patient healthcare interactions make this database well suited for evaluation of national trends in influenza vaccination for the overall population and among different subgroups.

This study used deidentified data in accordance with the Health Insurance Portability and Accountability Act (HIPAA) Privacy Rule deidentification standards and was conducted and reported following the International Society for Pharmacoepidemiology Guidelines for Good Pharmacoepidemiological Practice International Society for Pharmacoepidemiology [8].

### 2.3. Included Population

Part 1 included all individuals in the Linked Dataset at 1 year of age or older at the start (August 1) of each influenza season who had EMR activity at least once during any of the six previous influenza vaccination seasons from 2015–2016 through 2020–2021. Each influenza vaccination season was defined as August 1 of one calendar year through January 31 of the next calendar year, supported by our data indicating that only 0.4% to 1.6% of individuals were vaccinated between February and May each season (Appendix A). Included individuals also had to have at least one visit recorded in the EMR in the 12 months before the start of the vaccination season and were continuously enrolled (as indicated by the enrollment status in the claims data) from the start of the baseline period (12-month period before the start of the influenza vaccination season) through the end of the vaccination season. 

For the multi-season cohort in Part 2, we included individuals 5 years of age or older who had activity recorded in the EMR at least once during each of the five influenza vaccination seasons (2015–2016, 2016–2017, 2017–2018, 2018–2019, and 2019–2020). In addition, included individuals had at least one EMR-recorded healthcare visit at least 12 months before the start of the 2020–2021 influenza vaccination season and were continuously enrolled in the claims dataset for the duration of all six vaccination seasons (i.e., from 1 August 2015, to 31 January 2021). 

For both Part 1 and Part 2 cohorts, individuals were excluded if they had missing sex or geographic data, if they were aged ≥9 years with more than one influenza vaccine record or aged <9 years with more than two influenza vaccine records during any influenza season, and if they had received more than one influenza vaccine type during any vaccination season. To comply with the HIPAA certification of the Linked Dataset as statistically deidentified, age was capped at 89 as of the year 2021 in the data set derived for this study. For example, if a patient was 95 in the year 2021, their birth year was set to 1932, making them 89 in 2021, 88 in 2020, etc., through to age 84 in 2016 and 83 in 2015. Thus, it was not possible for subjects to be aged ≥85 years in the 2015–2016 or 2016–2017 influenza seasons.

### 2.4. Measures and Groupings

Vaccinations and patient characteristics were identified in the Linked Dataset through International Classification of Diseases, 9th Edition, Clinical Modification (ICD-9-CM) and International Classification of Diseases, 10th Edition, Clinical Modification (ICD-10-CM) diagnosis codes, Vaccine Administered codes (CVX, Appendix A), Current Procedural Terminology codes, and National Drug Codes. Age bands of 1 to 4, 5 to 12, 13 to 17, 18 to 49, 50 to 64 and ≥65 years were used in the analysis. Additional subgroupings in those >65 years included 65 to 69, 65 to 74, 75 to 84, and ≥85 years of age. Individuals at high risk of influenza complications had ≥1 EMR or medical claim with a diagnosis code for one or more medical conditions known to increase a person’s risk of experiencing serious influenza complications (Appendix A). The Charlson Comorbidity Index (CCI) [9] was used to measure the burden of comorbid conditions. 

For Part 2, vaccination history was determined via data from the five influenza seasons before 2020–2021, namely, the 2015–2016 through 2019–2020 seasons and grouped into 6 vaccination history patterns (Appendix A). The *consistently vaccinated* cohort included individuals who had received the influenza vaccination during all of five previous seasons. The *previous adopters* cohort included those who had received an influenza vaccine during 2 to 4 consecutive seasons before 2020–2021. *Previous adopters* included individuals who were vaccinated (i) yearly for 4 seasons (2016–2017 through 2019–2020); (ii) yearly for 3 seasons (2017–2018 through 2019–2020); or (iii) in both the 2018–2019 and 2019–2020 seasons. The cohort *vaccinated 2019–2020 only* comprised individuals who received influenza vaccine only during the 2019–2020 vaccination season but not in any of the prior seasons. The *not vaccinated* cohort comprised individuals who had not received an influenza vaccine during any of the influenza vaccination seasons before 2020–2021. Finally, the *inconsistently vaccinated* cohort included all other participants.

### 2.5. Statistical Analysis

For descriptive analyses, continuous variables were summarized using means and standard deviations as well as medians and interquartile ranges where appropriate. Categorical variables were summarized using counts and percentages. For Part 1, results were stratified by vaccination season. For Part 2, results were stratified by five-season vaccination history categories and 2020–2021 vaccination status. Standardized mean differences were used to assess potential differences between 2020 and 2021 influenza vaccinated vs. unvaccinated individuals. Standardized mean differences were used because they were not affected by sample size and can be used to compare the balance in measured variables between vaccinated and unvaccinated individuals. 

## 3. Results

### 3.1. Influenza Vaccination Trends, 2016–2017 through 2020–2021

Influenza vaccine uptake was highest in the 2020–2021 season (35.4%) during the COVID-19 pandemic (Figure 1). Between 2.37 to 2.87 million individuals in each influenza season (2016–2017 through 2020–2021) were included for Part 1 to describe influenza vaccine uptake patterns (Appendix A). For the 2015–2016 season, only 749,678 patients met all selection criteria because the medical and pharmacy claims data feeds began in 2015. Because of the fewer data contributors, the 2015–2016 season may not be directly comparable to later seasons and was not included in determining trends in Part 1. 

Appendix A describes the demographic and clinical characteristics of the total cohort (vaccinated and non-vaccinated) during each season (for part 1). Across the 2016–2017 through 2020–2021 seasons, the included populations shifted toward adults ≥65 years in the later seasons, with the proportion increasing from 17.7% in 2016–2017 to 24.9% in 2020–2021 (Appendix A). In contrast, the proportion of individuals aged 1 to 17 years decreased from 20.7% in 2016–2017 to 14.2% in 2020–2021. Hypertension was the most common comorbidity (approximately one-third of individuals each season, Appendix A), followed by type 2 diabetes and chronic pulmonary disease (approximately 15% each season). Chronic conditions that are common with increasing age, such as cardiovascular diseases and diabetes, increased slightly from the 2016–2017 to 2020–2021 seasons, reflecting the uptrend in age across the seasons.

### 3.2. Vaccination Trends in Subgroups

Influenza vaccine uptake increased between the 2016–2017 (31.4%) and 2020–2021 seasons (35.4%) (Figure 1). Across all seasons, the 18–49 age group consistently had the lowest vaccine uptake, while the oldest (age ≥ 65) and youngest (ages 1–4) age groups had the highest uptake (Figure 1). Vaccine uptake slightly increased since 2016–17 from season to season in all age groups except for the ≥65 age group, where it decreased slightly between the 2018–2019 and 2019–2020 seasons (from 47.5% to 45.0%) and stayed roughly the same in the 2020–2021 season (44.8%). In the 2020–2021 season, vaccine uptake increased slightly for patients 65 to 74 years of age (from 43.8% to 44.0%) but decreased in patients ≥75 years of age (from 47.3% to 46.6% in those 75–84 years from and 45.0% to 43.9% in individuals ≥85 years of age). Among children and working-age adults, influenza vaccine uptake increased steadily over the past vaccination seasons from 2016–2017 to 2020–2021, with a steeper increase observed in the 2020–2021 season for those <18 years of age.

There were differences in vaccine uptake among race and risk subgroups. Influenza vaccine uptake was highest among Asian and lowest among Black individuals (Figure 2). High-risk individuals were also more likely to be vaccinated than low-risk individuals, and these differences were more pronounced among adults than among children (Figure 3A,B). In the 1–4 and 5–12 age groups during the 2020–2021 season, a higher proportion of low-risk individuals received the influenza vaccine compared with high-risk individuals; however, the magnitude of the difference (2.0% in the 1–4 age group and 0.7% in the 5–12 age group) was small. 

### 3.3. Vaccination Timing and Settings

October was the peak month for vaccinations in all influenza seasons (Appendix A). In the 2020–2021 influenza season, 78.1% of vaccinated individuals received their influenza vaccinations between August and October (early vaccinations), compared with only 64.2% to 68.9% in the previous five seasons (Appendix A). 

The proportion of patients identified as being vaccinated via pharmacy claims was highest in the 2020–2021 season (36.0%) and was consistent with the increasing trend of pharmacy influenza vaccine claims over the previous seasons (from 15.7% in 2016–2017 to 28.3% in 2019–2020). A concomitant decrease in office/outpatient-based administration was also observed across the seasons. However, it is possible that not all vaccinations identified via pharmacy claims were administered in a pharmacy. For example, a vaccine could have been dispensed from a pharmacy and administered at an outpatient clinic. Influenza vaccinations in the hospital or other settings accounted for ≤1% of the total vaccinations in each season. 

### 3.4. Geographic Differences in Influenza Vaccination Uptake

Between the 2019–2020 and 2020–2021 influenza seasons, 32 (62.7%) of the 50 states (plus the District of Columbia) saw an increase in vaccination uptake (Appendix A). The largest increases were in Vermont (11.3% increase), Massachusetts (8.4% increase), and California (4.9% increase). Those with the largest decreases were Idaho (7.2% decrease), West Virginia (5.7% decrease), and Montana (5.6% decrease). Considering a broader time span, between the 2016–2017 and 2020–2021 seasons, 47 (92.2%) of the 50 states (plus the District of Columbia) had increases in influenza vaccine uptake.

### 3.5. Individual Characteristics According to Influenza Vaccination History from the 2015–2016 through the 2019–2020 Seasons

Of 9,519,492 patients enrolled in the data set during the 2020–2021 season, 325,389 met all the inclusion criteria for Part 2 and were included in the analysis of vaccination history (Appendix A). The mean (SD) age of this continuously enrolled cohort was 55.0 (22.9) years among vaccinated individuals and 54.3 (20.5) years among unvaccinated individuals in 2020–2021 season (Appendix A). The age of this cohort was higher than that of each of the prior influenza seasons (ranging from 43 to 47 years, Appendix A). A higher proportion of patients were vaccinated in the 2020–2021 season in the continuously enrolled population for Part 2 (42.7%) compared with the population for Part 1 (35.4%, Figure 1), which may reflect the higher age and/or increased interactions with healthcare providers in this continuously enrolled cohort. Within the Part 2 cohort, only 13% (n = 43,140) were *consistently vaccinated* during the previous five influenza seasons, 30% were *previous adopters* (n = 97,202), 17% were *inconsistently vaccinated* (n = 54,629), 5% were *vaccinated 2019–20 only* (n = 14,937) and 35% were *not vaccinated* during any of the previous five seasons (n = 115,481) (Appendix A). 

Children <18 years and older adults ≥65 years were most likely to be consistently vaccinated in each of the previous five influenza seasons (*consistently vaccinated* cohort), while the 18–49 and 50–64 age groups were most likely to be not vaccinated in any of the previous influenza seasons (*not vaccinated* cohort, Figure 4, Appendix A). Additionally, a higher proportion of White (14.8%) and Asian (17.7%) patients were in the *consistently vaccinated* cohort compared with Black patients (7.5%). Conversely, a larger proportion of Black patients (43.2%) were in the *not vaccinated* cohort compared with White (33.2%) and Asian (27.0%) patients (Figure 5, Appendix A). Individuals in the *not vaccinated* cohort also had the lowest CCI score (i.e., fewer comorbidities) compared with other vaccination history cohorts (Appendix A). A higher frequency of chronic pulmonary disease (18.5% vs. 13.8%), diabetes with chronic complications (11.2% vs. 7.5%), peripheral vascular disease (11.2% vs. 7.8%), and renal disease (8.0% vs. 5.3%) were observed in the *consistently vaccinated* cohort compared with the *not vaccinated* cohort.

### 3.6. Influenza Vaccination Status in the 2020–2021 Season Based on 5-year Vaccination History

The study population for part 2 included 138,824 vaccinated individuals (42.7%) and 186,565 unvaccinated individuals (57.3%) in the 2020–2021 influenza season (Appendix A). Only the youngest age group (5 to 12 years) had more vaccinated individuals (57.4%) than unvaccinated individuals for the 2020–2021 season. Individuals 18 to 49 years of age had the lowest 2020–2021 vaccination uptake (15.2%). Influenza vaccination status in the 2020–2021 season was generally consistent with previous vaccination history. Individuals in the *not vaccinated* cohort were least likely, and those in the *consistently vaccinated* cohort were most likely, to be vaccinated in 2020–2021 (Figure 6). 

Few differences in patient characteristics were observed in the 2020–2021 influenza season between vaccinated vs. unvaccinated individuals when stratified by vaccination history (Appendix A). In all vaccination history cohorts except the *not vaccinated* cohort, a higher proportion of 18-to-49-year-olds were unvaccinated vs. vaccinated in 2020–2021 (standardized mean differences > 0.1): 12.4% vs. 9.3%, in the *consistently vaccinated* cohort; 18.1% vs. 14.3% in *previous adopters*; 25.3% vs. 20.7% in the *inconsistently vaccinated* cohort; and 27.5% vs. 20.4% in the *vaccinated 2019–2020 only* cohort. The higher proportion of 18-to-49-year-olds unvaccinated in 2020–2021 was consistent with the data from part 1 showing that 18-to-49-year-olds consistently had the lowest influenza vaccine uptake across seasons. 

Differences other than age were observed between the 2020–2021 vaccinated and unvaccinated individuals, mostly in those who had been *consistently vaccinated*. For example, in those who were *consistently vaccinated*, 2020–2021 vaccine recipients had a higher proportion of 5-to-12-year-olds, a lower mean (SD) CCI score (1.2 [1.7] vs. 1.4 [2.0]), lower rates of peripheral vascular disease and dementia, and a lower proportion of all-cause emergency department (ED) visits in the preceding 12 months (16.7% vs. 20.9%), compared with 2020–2021 unvaccinated individuals (Appendix A). Furthermore, in the *consistently vaccinated* cohort in the 2020–2021 influenza season, there was a higher proportion of Pennsylvania residents (18.2% vs. 13.1%) and a lower proportion of Tennessee residents (3.9% vs. 8.4%) in the vaccinated group than in the unvaccinated group. In the *inconsistently vaccinated* and *not vaccinated* cohorts in the 2020–2021 influenza season, there were higher proportions of California residents in vaccinated in 2020–2021 than unvaccinated (17.0% vs. 13.0% and 17.2% vs. 13.1%, respectively).

## 4. Discussion

The highest influenza vaccine uptake was observed in the 2020–2021 season (35.4%) during the COVID-19 pandemic and was consistent with the trend of increasing vaccine uptake observed since the 2016–2017 season (31.4%). While the CDC has also reported that uptake of influenza vaccination is improving, the estimates from the current study were consistently lower than the CDC’s estimates every season (which were 46.8% in 2016–2017 and 52.1% in 2020–2021) [10,11]. CDC estimates were based on telephone survey data [10], whereas our analysis used data reported in medical records and claims and may not have captured an estimated 15% of vaccinations administered at workplaces, health departments, and other public facilities [10]. Additionally, recall errors and inherent differences between survey respondents and non-respondents may have led to overestimation of vaccination uptake collected through surveys. Despite differences between our claims/EMR-based data and CDC’s survey data, trends observed in this study generally mirror those reported by the CDC [10,11]. Similar to CDC estimates, we observed that vaccine uptake was lowest in the 18–49 age group and highest in the oldest age group (age ≥ 65) and youngest age group (ages 1–4).

Increased post-pandemic (2020–2021) influenza vaccination coverage from prepandemic (2019–2020) levels among individuals older than 65 years has also been reported for several countries in both the Northern and Southern hemispheres (with between 3.3% to 13.0% relative increases) [12]. although South Korea experienced a 10.2% decrease in coverage over the same period [12]. Vaccination coverage in Shanghai, China, increased from 10.8% in 2017–2018 to 50.8% in 2020–2021 [13].

Our observation of earlier vaccine uptake in 2020–2021 compared to prior seasons was an encouraging trend. In preliminary influenza vaccination uptake estimates, the CDC found that by mid-January 2021, the highest number of influenza vaccine doses were distributed in a single season, with 53% of adults vaccinated by mid-January 2021, compared with 45% at the end of January 2020 [14]. Additionally, in concurrence with CDC estimates of worsening racial disparities in influenza vaccine uptake [15], we show that during the COVID-19 pandemic in 2020–2021, vaccine uptake was lowest among Black individuals compared with Asian and White individuals. The COVID-19 pandemic may have exacerbated inequities in education, income, paid leave, and access to healthcare services [16]. Vaccine misinformation may have furthered distrust in the healthcare system and contributed to a higher perceived vaccine risk and greater vaccine hesitancy among minorities [17,18]; lack of information on vaccine benefits and current vaccine recommendations also reduced confidence in the vaccine [19]. Because influenza-related hospitalization rates are highest among minorities [15], closing the gap in racial disparities in influenza vaccination may substantially alleviate the burden of influenza on patients’ health as well as stress on the medical system.

During the 2020–2021 season, influenza vaccine uptake across age groups was higher for high-risk individuals compared with low-risk individuals, except for children ≤12 years of age. In children ≤12 years, we observed that vaccine uptake also increased for those with low underlying risk. This finding contrasts with reports of a decline in the pediatric influenza vaccination rate [10,20,21] as well as in routine childhood vaccinations [22,23,24] during the COVID-19 pandemic. These previous reports based on data from household surveys, state immunization information systems, and individual healthcare systems may have included uninsured and insured participants. Previous data suggested that vaccine uptake for influenza and other preventive vaccines is lower among uninsured populations [25]. Our findings of an increased rate of influenza vaccine participation in low-risk pediatric patients during the 2020–2021 influenza season may reflect better vaccine access in our insured pediatric population.

Only a small proportion (13.2%) of continuously enrolled individuals was consistently vaccinated in all five of the most recent influenza seasons (2015–2016 through 2019–2020). Similar to the cohort in Part 1, White and Asian individuals, children, and individuals with more comorbidities were more likely to be consistently vaccinated, whereas individuals 18 to 49 years of age were more likely to be not vaccinated in any of the previous five seasons. The more consistent vaccination rate in children may be due to the universal coverage of vaccines for children, even for uninsured children, through the Vaccines For Children program [26]. 

Influenza vaccination status in the 2020–2021 season was generally consistent with previous vaccination history. Individuals who were consistently vaccinated in the previous five seasons were also most likely (83.7%) to be vaccinated in the 2020–2021 season, whereas patients who were not vaccinated in any of the previous five seasons were least likely (14.2%) to be vaccinated for the 2020–2021 season. Other researchers have reported that previous vaccination against influenza has been associated with future influenza vaccine uptake [5,6,7]. In our report, 52% of continuously enrolled individuals were previously vaccinated at least some of the time, which may be the “low hanging fruit” to improve population vaccination uptake [27]. Even if this group lacks strong motivation for annual influenza vaccinations, improving access and providing reminders to heighten awareness may be effective in increasing the vaccination rate [27].

Overall, when stratified by vaccination history, there were few differences in demographic or clinical characteristics between those who were vaccinated in the 2020–2021 season compared with those who were not. Our findings suggest that within vaccination history groups, the decision to receive the influenza vaccine in a subsequent season may be primarily driven by factors other than demographic or clinical characteristics, and these factors were not assessed in our study. These unmeasured factors may include patients’ perceived risk of vaccination compared with influenza infection, perceived risk of seeking healthcare during a pandemic [28], access to healthcare, and socioeconomic status [16].

Contrary to expectations, individuals in the *consistently vaccinated* cohort who were vaccinated in the 2020–2021 season were younger, had lower mean CCI scores, and had fewer all-cause ED visits in the preceding 12 months than those who were unvaccinated in the 2020–2021 season. Our observation of lower vaccination uptake in the >65-year and high-risk groups is concordant with findings shown in season-to-season vaccination trends, with a stagnant vaccination rate between the 2019–2020 and 2020–2021 seasons for individuals ≥65 years. CDC data also showed a reduction in the influenza vaccination rate in people aged 18 to 64 years at high risk in the 2020–2021 season compared with the 2019–2020 season [11]. The lower vaccine uptake among these groups in the 2020–2021 season may have been due to deferral of care for chronic medical conditions and preventive care during the COVID-19 pandemic [29]. A survey study reported that 30% of primary care, 50% of internal medicine, and 57% of cardiovascular patients deferred care during the pandemic; deferral of preventive care was even more likely, with 75%, 65%, and 86% of these patients reporting delaying care, respectively [29]. Older adults may also have had difficulty adapting to changing health system structures during the pandemic [30], and the inability of caregivers to visit due to physical distancing may have created challenges for older adults to attend their healthcare and vaccination appointments [30]. Recognizing the challenges that physical distancing created for vulnerable older adults may inform future vaccination programs during pandemics.

This study has several strengths. First, our sample size is large, with an average of 2.34 million individuals per influenza season in Part 1 for the evaluation of season-to-season vaccination trends and 325,389 individuals in Part 2 for the evaluation of vaccination history. Additionally, the linked claims and EMR dataset comprised healthcare practices covering every 3-digit zip code in the United States. The large sample size and broad geographic coverage minimized regional differences in the included population and provided more generalizable findings. Moreover, our observations across six influenza seasons allowed for the evaluation of influenza vaccine history. 

Our study also has several limitations. Individuals may have received vaccinations outside of a healthcare setting, which may have led to an underestimation of vaccine uptake within the database. However, vaccinations outside healthcare settings are not expected to affect our analysis of trends in vaccine uptake. The inclusion criteria requirements for EMRs and claims activity may have limited our population of healthy individuals who were vaccinated against influenza but had no other EMR activities. The requirement of continuous enrollment in the database for five seasons for determination of vaccination history resulted in large decreases in cohort sizes, potentially decreasing the generalizability of our findings from the vaccination history cohorts. Finally, because the medical and pharmacy claims data feeds began in 2015, the 2015–2016 influenza season had a smaller included population and may not be directly comparable to later seasons. 

## 5. Conclusions

Overall influenza vaccine uptake was highest in 2020–2021 during the COVID-19 pandemic, following an increasing trend since 2015–2016. Differences in vaccine uptake among age, race, and risk-status subgroups as well as states of residence persisted over the seasons. A higher proportion of patients were vaccinated before November in the 2020–2021 season compared with past seasons. Only a small portion of patients were consistently vaccinated across seasons. Children ≤17 years of age and adults ≥65 years of age as well as Asian and White individuals were most likely to be consistently vaccinated. Individuals 18 to 49 years of age and those with fewer comorbidities were least likely to be consistently vaccinated. Despite being consistently vaccinated in the previous five influenza seasons, older individuals, those with more comorbidities, and individuals with a history of higher healthcare resource utilization were less likely to receive the influenza vaccine in the 2020–2021 season. However, an increase in vaccine uptake was observed in the pediatric low-risk group. Few demographic and clinical characteristics differed between 2020–2021 vaccine recipients vs. nonrecipients when stratified by vaccination history. 

This study adds to existing literature by providing an assessment of influenza vaccine uptake trends using integrated EMR and claims data in contrast to the standard approaches of surveys and registries, which allowed evaluation of evaluation of multiseason vaccination history trends. Understanding disparities in influenza vaccine uptake among different subgroups can help direct public health efforts to increase vaccine coverage. Future research should evaluate additional factors (e.g., individual, sociodemographic, and contextual factors) associated with influenza vaccine uptake, as well as ways to improve influenza vaccine access for the elderly and high-risk individuals during a pandemic. 

## Figures and Tables

**Figure 1 vaccines-10-01610-f001:**
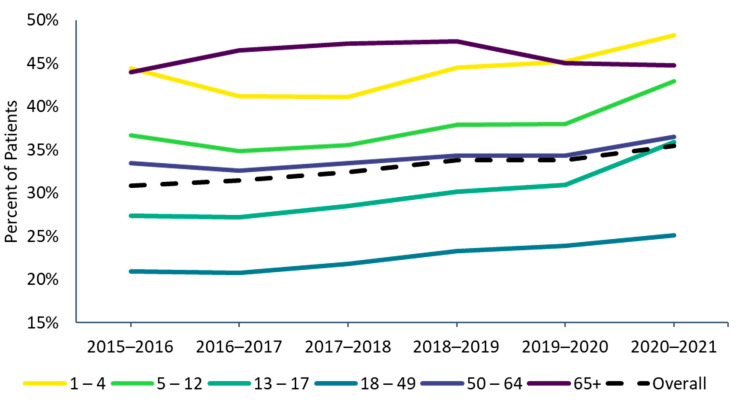
Influenza vaccine uptake by age group, 2015–2016 through 2020–2021 influenza seasons. ^1^ The 2015–2016 season was not included in the trend analyses in Part 1. Because the medical and pharmacy claims data feeds began in 2015, the 2015–2016 season had fewer data contributors and may not be directly comparable to later seasons.

**Figure 2 vaccines-10-01610-f002:**
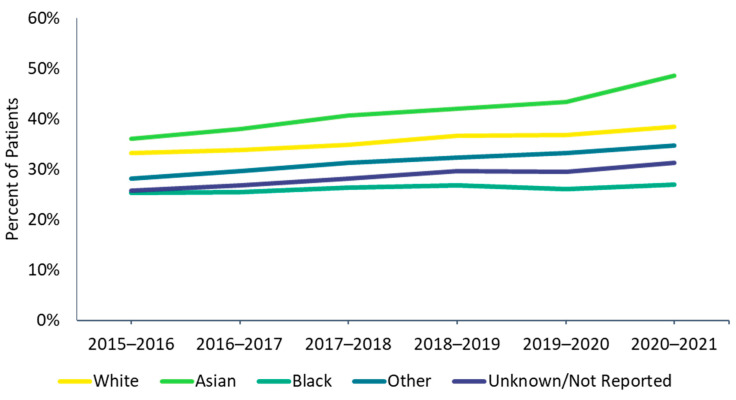
Influenza vaccine uptake by race, 2015–2016 through 2020–2021 influenza seasons.

**Figure 3 vaccines-10-01610-f003:**
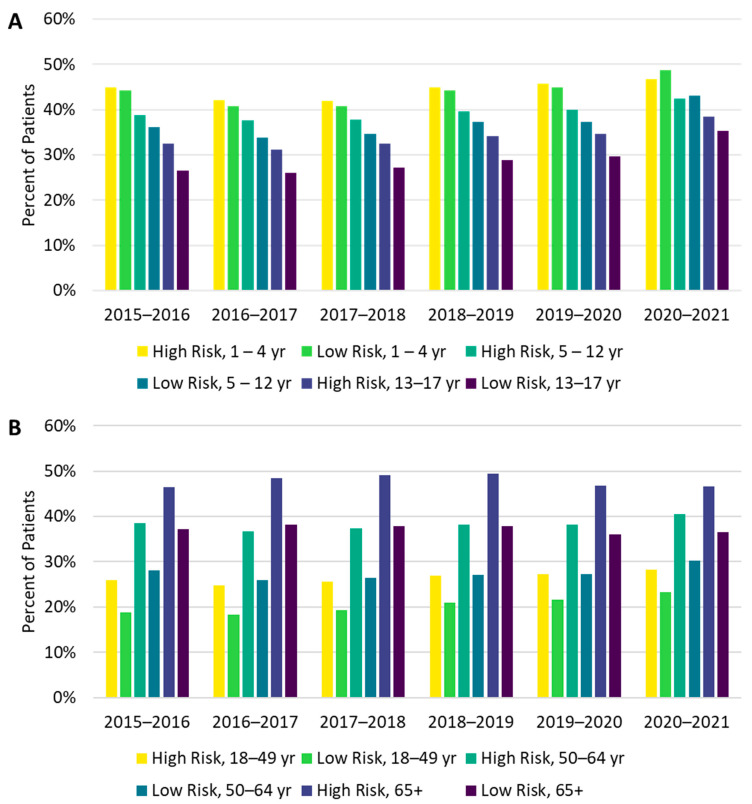
Vaccine uptake from the 2015–2016 through the 2020–2021 influenza seasons by risk and age group for individuals (**A**) 1–17 years of age and (**B**) 18 years and older.

**Figure 4 vaccines-10-01610-f004:**
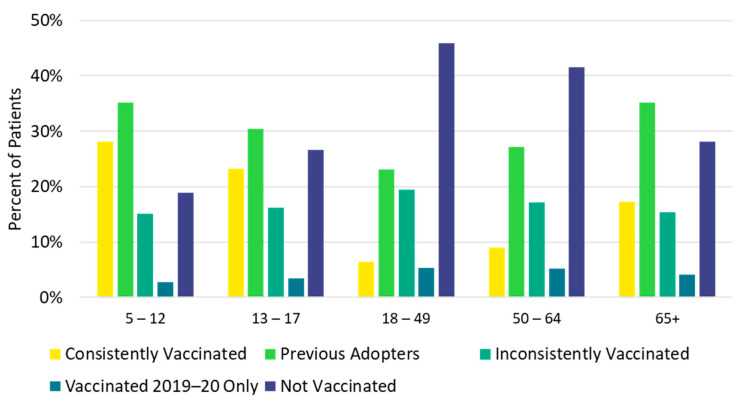
Vaccination history by age group: 2015–2016 through 2019–2020.

**Figure 5 vaccines-10-01610-f005:**
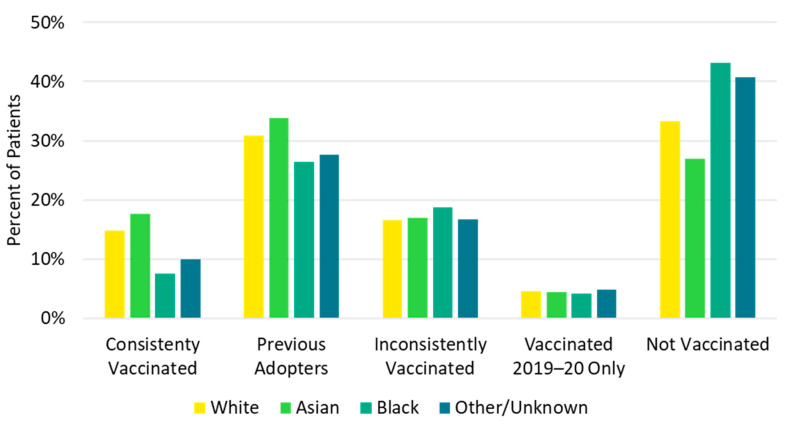
Vaccination history by race.

**Figure 6 vaccines-10-01610-f006:**
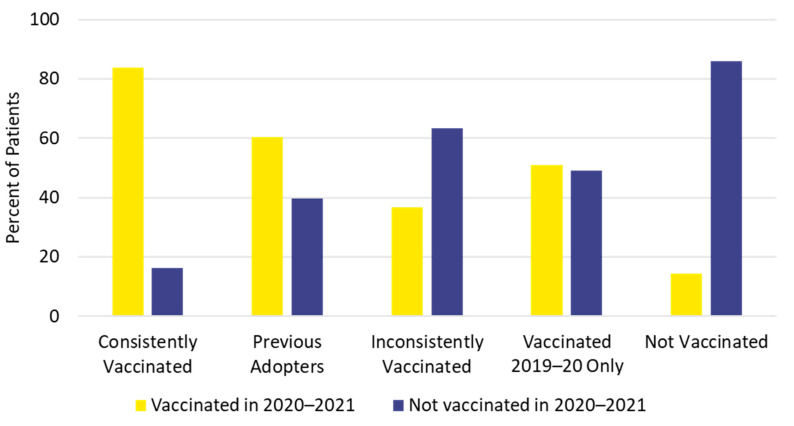
Influenza vaccination history and vaccination status for the 2020–2021 season.

## Data Availability

The datasets used in this study are privately licensed and are not available to be shared publicly.

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
