# Peer review of "Influenza Vaccine Uptake in the United States before and during the COVID-19 Pandemic"

_vaccines, 2022, doi:10.3390/vaccines10101610_

Round 1

Reviewer 1 Report

Thank you very much for the invitation. This is an interesting study but I cannot fully evaluate the manuscript without complete information. The manuscript states many supplementary tables and figures, but I am unable to track them during my review. Please provide the complete details on the manuscript so I could see the analysis. I suggest the authors consider inferential statistics to analyze the data. Most of the results are presented in a descriptive way. A regression analysis of the factors associated with vaccine uptake before and after the covid19 may reveal more useful information. 

Author Response

Author Response to Reviewer Comments

Manuscript Number: vaccines-19805130

Manuscript title: Influenza vaccine uptake in the United States before and during the
COVID-19 pandemic
Authors: Ian McGovern, Alina Bogdanov, Katherine Cappell, Sam Whipple, Mendel Haag

Reviewer #1

  • Thank you very much for the invitation. This is an interesting study but I cannot fully evaluate the manuscript without complete information. The manuscript states many supplementary tables and figures, but I am unable to track them during my review. Please provide the complete details on the manuscript so I could see the analysis.  

Response: We have now provided complete information, including all the supplementary tables and figures.

  • I suggest the authors consider inferential statistics to analyze the data. Most of the results are presented in a descriptive way. A regression analysis of the factors associated with vaccine uptake before and after the covid19 may reveal more useful information. 

Response:  Thank you for your comment. We agree that inferential statistics can be helpful in identifying key drivers of vaccine uptake. However, in this instance there was a deliberate decision to focus on descriptive statistics because univariate coverage statistics are a more common method of describing how (influenza) vaccine coverage may vary among different subgroups. Absolute differences in the proportion of the populations vaccinated may resonate better with a general audience than a measure of how odds of vaccination differ. Additionally, univariate statistics allow for comparisons across published literature in terms of how coverage levels differ across different data sources/seasons/regions. Finally, it is not anticipated that in most instances that adjusting for other factors included would meaningfully clarify the trends seen from the descriptive analysis. For example, adjusting for factors like age, sex, and risk status is unlikely to change the observed trend of racial disparities in coverage.

Reviewer 2 Report

Thank you for giving me the opportunity to review this manuscript by McGovern and colleagues. It is an interesting paper aimed at describing influenza vaccine uptake patterns before and during the onset of the pandemic in an insured population; moreover, it aims to describe end evaluate the differences between different populations based on their vaccination history. 

Congratulations with the manuscript, which is well-written. One major issue and some minor issues before it can be published: 

- in the manuscript, the authors have specified that ethical review and approval were waived because the study analyzes de-identified data from an electronic database. Despite presenting anonymous/aggregate data, access to personal data was required to build the study: therefore, based on my understanding on what has been written on the paper, the ethic approval should be requested (even if the study is conducted in accordance with the HIPAA, which is a requirement for ethic approval, not an alternative). If my interpretation is wrong, I suggest expliciting - and making it clearer - the reasons why the ethic approval could be waived. 

- In the data source section, I suggest better describing the database used, and explaining why this specific database has been used. 

- In the discussion section: please revise the first paragraph (290-303) as the central part is hard to follow.

- The discussion section is quite long (I suggest shortening it); however, I think that two important things are missing -> a brief confrontation of your findings with coverage rates in different countries in the same period ([1-2] find below some examples) and one explicit paragraph about the public health implications of your important findings (many Vaccines readers are Public health experts and may benefit from this). 

1. https://www.ncbi.nlm.nih.gov/pmc/articles/PMC8728488/. 

2. https://pubmed.ncbi.nlm.nih.gov/35621293/

Author Response

Author Response to Reviewer Comments

Manuscript Number: vaccines-19805130

Manuscript title: Influenza vaccine uptake in the United States before and during the
COVID-19 pandemic
Authors: Ian McGovern, Alina Bogdanov, Katherine Cappell, Sam Whipple, Mendel Haag

Reviewer #2

  • In the manuscript, the authors have specified that ethical review and approval were waived because the study analyzes de-identified data from an electronic database. Despite presenting anonymous/aggregate data, access to personal data was required to build the study: therefore, based on my understanding on what has been written on the paper, the ethic approval should be requested (even if the study is conducted in accordance with the HIPAA, which is a requirement for ethic approval, not an alternative). If my interpretation is wrong, I suggest explicating-and making it clearer-the reasons why the ethic approval could be waived. 

Response: Thank you for your comment. The data utilized for this analysis went through a deidentification process before access for study purposes. The deidentified data was certified via independent external expert determination (specifically by Mirador analytics) to not contain sufficient information to identify individual patients. Because the data utilized met HIPAA privacy rule deidentification standards, no individual consent or ethics approval is required. We have edited the manuscript to clarify that the mention of HIPAA was specifically in reference to compliance to HIPAA deidentification standards. DHHS provides a more complete description of this process and applicable legislation here: https://www.hhs.gov/hipaa/for-professionals/privacy/special-topics/de-identification/index.html

  • In the data source section, I suggest better describing the database used, and explaining why this specific database has been used. 

Response:  Thank you for your comment. In the data source section we have clarified that the use of closed claims from the payer provides a comprehensive view of patient interactions and added the following rationale for the suitability of the database: “The combination of national representativeness, large sample size, and comprehensive insights into patient healthcare interactions make this database well suited for evaluation of national trends in influenza vaccination for the overall population and among different subgroups.”

  • In the discussion section: please revise the first paragraph (290-303) as the central part is hard to follow

Response: We have revised the text to make it clearer

  • The discussion section is quite long (I suggest shortening it)

Response: Thank you for the suggestion. We have shortened the discussion section by about one third by revising the text to be more concise and by focusing on the most important components of the discussion.

  • I think that two important things are missing: a brief confrontation of your findings with coverage rates in different countries in the same period ([1-2] find below some examples) and one explicit paragraph about the public health implications of your important findings (many Vaccines readers are Public health experts and may benefit from this). 

Response: Thank you for the suggestion. We have added text that compares our findings with those from other countries

The following text about the public health implications has been added to the conclusion section: “This study adds to existing literature by providing an assessment of influenza vaccine uptake trends using integrated EMR and claims data in contrast to the standard approaches of surveys and registries, which allowed evaluation of evaluation of multiseason vaccination history trends. Understanding disparities in influenza vaccine uptake among different subgroups can help direct public health efforts to increase vaccine coverage.”

Round 2

Reviewer 1 Report

Thank you for your response. As I suggested further analysis, the authors clarified the factors hindering them to do further analyses. I agreed with them.

Reviewer 2 Report

Thank you for providing a new version of the manuscript, addressing all my comments and giving explanations. I have no further suggestions.

Best regards